# Efficacy and safety of PM-AR-T versus edwards MC3 rings in tricuspid regurgitation: A non-inferiority, randomized controlled trial

Zhenjun Xu[1☯], Jie Li[2☯], Wanzi Xu[1], Qiuyan Zong[1], Wenjie Ji[1], Yin Xu[1], Yunyan Su[1], Kunsheng Li[1], Dongjing Wang 🔾[1]*, Jun Pan[1]*

1 Department of Cardiothoracic Surgery, Nanjing Drum Tower Hospital, The Affiliated Hospital of Nanjing University Medical School, Nanjing, China, 2 Department of Ultrasound Imaging, Surgery, Nanjing Drum Tower Hospital, The Affiliated Hospital of Nanjing University Medical School, Nanjing, China

☯ These authors contributed equally to this work.
* pj791028@163.com (JP); glwdj@nju.edu.cn (DW)

## Abstract

### Objectives

Tricuspid valve repair, particularly with annuloplasty rings, is increasingly recognized as an effective treatment. PM-AR-T is a semi-rigid annuloplasty ring based on a nickel-titanium alloy which has made progress in animal models, however, studies on PM-AR-T's performance in patients with tricuspid regurgitation (TR) are lacking. This study aimed to compare the efficacy and safety of the PM-AR-T with the Edwards MC3 ring for the TR treatment.

### Methods

A non-inferiority, randomized controlled trial was conducted in 20 centers across China, enrolling patients with tricuspid valve disease requiring surgical repair. Patients were randomized to receive either PM-AR-T or Edwards MC3 ring. The primary endpoint was the success rate of valve repair at 6 months.

### Results

A total of 164 patients underwent valve annuloplasty, 83 and 81 in the PM-AR-T and Edwards MC3 groups. Valve repair success rates were 92.8% and 93.8% in the PM-AR-T and Edwards MC3 groups, demonstrating non-inferiority with a difference of −1.1% (95% confidence interval [CI]: −9.5 to 7.4), which was less than the pre-specified non-inferiority margin of −10%. No significant intergroup differences were found in valve regurgitation, echocardiographic parameters, and New York Heart Association (NYHA) functional classification at any postoperative time point. At 12 months, the proportions of patients without regurgitation were comparable, 30.4% and 27.8% in the PM-AR-T and Edwards MC3 groups (P = 0.705). Improvement to

**Data availability statement:** All relevant data are within the manuscript and its Supporting Information files.

**Funding:** Beijing Permed Biomedical Engineering Co., Ltd.

**Competing interests:** The authors have declared that no competing interests exist.

NYHA functional class I status was detected in 46.2% and 45.6% of the two groups by 12 months (P = 0.893). Both rings exhibited comparable safety profiles, with no device-related serious adverse events, cardiovascular deaths, major bleeding events, severe structural damage, infective endocarditis, or thromboembolic events.

## Conclusions

The PM-AR-T tricuspid valve semi-rigid ring is effective in improving TR, demonstrating non-inferiority to the Edwards MC3 ring, with a favorable safety profile.
**Clinical Trial Registration:** This study was registered at Chinese Clinical Trial Registry (ChiCTR2100043007).

## Introduction

Tricuspid regurgitation (TR) represents a significant clinical challenge within the spectrum of valvular heart diseases [1,2]. Characterized mostly by functional tricuspid regurgitation (FTR), which accounts for 70–80% of all TR cases, this condition is primarily induced by right ventricular dilation and annular dilatation [3]. In the current therapeutic landscape, management strategies for TR encompass two major approaches: tricuspid valve replacement and tricuspid valve repair [4,5]. Tricuspid valve repair, especially by prosthetic annuloplasty rings, is highly effective in restoring valve function and achieving anatomical correction, providing a significant advancement in the treatment modalities available for TR [6].

In the field of tricuspid valve repair, the prosthetic annuloplasty rings have two primary variants dominating current medical practice, including rigid and flexible rings [7,8]. Rigid rings are lauded for their ability to preserve annulus shape, effectively reducing annular dilation and curtailing regurgitation. However, their rigidity compromises the natural contractility and compliance of the annulus. Conversely, flexible rings offer enhanced adaptability to the dynamic movements of the tricuspid valve, although they are prone to deformation under prolonged stress conditions such as pulmonary arterial, which may lead to reduced control over regurgitation [9]. Edwards MC3 tricuspid annuloplasty ring, with its semi-rigid, 3D saddle-shaped design that closely mirrors the anatomical structure of the tricuspid valve annulus. Enhanced with a titanium alloy, this ring combines the rigidity required for effective shape maintenance with enough elasticity to minimize suture tension, aiming to reduce the odds of TR recurrence [10]. Preliminary analysis of the Edwards MC3 ring has demonstrated its efficacy in alleviating TR, alongside a favorable safety profile [11–13]. Similar to titanium alloy, materials such as nickel-titanium alloy, renowned for their superelasticity [14]. These materials, capable of enduring significant strain without permanent deformation, underscore the potential for tricuspid valve repair technologies that integrate with the physiological dynamics of the heart, ensuring both durability and functional integrity in the management of TR.

The PM-AR-T semi-rigid annuloplasty ring, an innovative product developed by Beijing PuHui Biomedical Engineering Co., Ltd., features a design that utilizes a 3D

saddle-shaped structure with a nickel-titanium alloy. The PM-AR-T incorporates a tubular lattice pattern filled with flexible wires. Compared to the solid rectangular cross-section of the material-matched Edwards MC3 ring, this design enhances freedom of motion at the septal portion, better conforming to native annular dynamics. Furthermore, the PM-AR-T features graded stiffness transition along the anterior-posterior (AP) axis, with maximal rigidity at the non-patterned segment within the AP region extending toward the opposite side. This design enables targeted correction of pathological dilation occurring along the AP direction. Preliminary studies in animal models, specifically small-tailed Han sheep, demonstrated comparable outcomes between PM-AR-T (n = 17) and Edwards MC3 rings (n = 12). Surgical mortality (PM-AR-T: 29.0%, Edwards MC3: 25.0%) was unrelated to devices. Among 21 survivors reaching endpoints (90 ± 7d: PM-AR-T, n = 5, Edwards MC3, n = 4; 180 ± 7d: PM-AR-T, n = 7, Edwards MC3, n = 5), echocardiography revealed no tricuspid regurgitation or thrombus in either group. Histopathology [15] indicated equivalent endothelialization, inflammatory responses, and biocompatibility. Gross pathology confirmed correct ring positioning without leaflet/coronary damage or device-related complications. Despite these promising insights, a gap remains in the understanding of PM-AR-T's performance in patients with TR.

To bridge this knowledge gap, a non-inferiority randomized controlled trial was performed to evaluate the efficacy and safety of PM-AR-T in the therapeutic management of TR.

## Methods

### Study design and patients

This prospective, multicenter, non-inferiority randomized controlled trial was performed in 20 centers across China, from February 8, 2021 to March 9, 2023.

Inclusion criteria were: 1) aged 18–75 years; 2) diagnosed with tricuspid valve disease attributable to degenerative or functional valve lesions, among other etiologies; 3) planned surgical repair; 4) New York Heart Association (NYHA) functional class III or lower.

Exclusion criteria were: 1) contraindications for the implantation of tricuspid valve annuloplasty rings or a history of tricuspid valve surgery; 2) NYHA functional class IV due to heart failure or current right heart failure; 3) intracardiac thrombus in either the right atrium or ventricle, as assessed by the investigator, potentially elevating the risk associated with surgery; 4) eligibility for heart transplantation surgery; 5) anticipated survival of less than one year; 6) cardiac surgery, interventional procedures, or cerebrovascular events in the three months preceding the study enrolment; 7) renal failure requiring dialysis; 8) severe respiratory disorders; 9) active infections; 10) contraindications to anticoagulant and antiplatelet medications; 11) documented allergies to materials utilized in the manufacturing of annuloplasty rings; 12) unsuitability for study inclusion as determined by the investigators.

This study was approved by the Nanjing Drum Tower Hospital (approval number 2020-291-02 and clinical trial number ChiCTR2100043007); registration date: February 4, 2021. Written consent to participate in the study was provided by all patients.

### Procedure

This study implemented a dynamic randomization technique, stratifying participants based on two criteria: treatment center and severity of regurgitation (categorized as mild, moderate, or severe). Randomization sequences were generated by the Clinical Trial Central Randomization System (DAS for IWRS), ensuring a 1:1 allocation of tricuspid valve repair patients to the PM-AR-T (PM-AR-T semi-rigid annuloplasty ring; Beijing PuHui Biomedical Engineering Co., Ltd.) or the Edwards MC3 (Edwards MC3 tricuspid valve annuloplasty ring; Edwards Lifesciences Shanghai Co., Ltd.) groups. The randomization process involved accessing the web-based system and inputting the required patient information. The system then returned the treatment assignment in real time, based on a pre-specified randomization algorithm incorporating stratification and dynamic allocation to ensure balance between groups.

All surgical interventions were performed under general anesthesia with cardiopulmonary bypass support. Upon induction of anesthesia, transesophageal echocardiography was performed to evaluate the tricuspid valve for pathology. Surgical access was gained through various incisions. Cardiopulmonary bypass was initiated via cannulation of the ascending aorta and both venae cavae, followed by induction of moderate hypothermia and pharmacological myocardial arrest to ensure cardiac protection. The tricuspid valve was accessed through a right atrium incision, and valve analysis and repair were guided by Carpentier's valve probing principles, with 2/0 Ethibond sutures placed intermittently around the annulus. Repair techniques were tailored to the specific valve pathology. The adequacy of repair was verified by saline infusion tests, assessing for proper ventricular filling. The choice of annuloplasty ring size was determined by measuring the intercommissural distance and the length of the anterior leaflet before and after these tests. Upon ring implantation, the efficacy of repair was reassessed with a second saline test, after which the right atrium incision was closed, and the patient was gradually weaned from bypass. Postoperative management included anticoagulation with heparin, transitioning to oral warfarin for three months, aiming to maintain an international normalized ratio between 2.0 and 3.0.

Intraoperative measurement of the tricuspid valve's septal base length served as the basis for selecting the suitable size and model of the prosthetic annuloplasty ring, with PM-AR-T or Edwards MC3 being selected referring to the initial randomization.

### Endpoints

The primary endpoint was the success rate of valve repair six months post-surgery. The success rate was defined by two criteria: non-occurrence of severe TR as verified by cardiac ultrasound, and postoperative increase in annular diameter not exceeding 15% [16,17]. This definition reflects established criteria used in previous studies [16,17]. Secondary endpoints included the degree of valve regurgitation, evaluated by echocardiography using the TR evaluation index; alterations in echocardiographic parameters (e.g., right atrial and ventricular dimensions, diameter and narrowest width of the regurgitant jet and area of regurgitant flow); modifications in the NYHA functional classification all recorded at hospital discharge or 30 days postoperatively (whichever came first) and then at 3, 6, and 12 months. Additionally, the study monitored the incidence of recurrent tricuspid valve surgery due to TR.

The safety endpoint was the serious adverse events (SAEs) and treatment-emergent adverse events (TEAEs), monitored at discharge or 30 days postoperatively (whichever came first) and then at 3, 6, and 12 months following the procedure. SAEs included 1) overall mortality rate (both cardiovascular and non-cardiovascular mortality rates); 2) device-related cardiovascular mortality, as defined by the Valve Academic Research Consortium (VARC) [18], encompassing myocardial infarction, pericardial tamponade, exacerbation of heart failure, and death associated with non-coronary vascular conditions or surgical complications; 3) device-related major bleeding, categorized as type 3b or higher based on the Bleeding Academic Research Consortium (BARC) criteria [19], excluding type 4 incidents; 4) incidents of device-related serious damage to cardiac or other body structures; 5) occurrences of device-related infective endocarditis; 6) and the incidence of device-related thromboembolic events, identified by clinical assessment or imaging indicative of thrombosis-induced vascular dysfunction [20].

An independent Clinical Endpoints Committee (CEC) reviewed all investigator-reported endpoints.

### Statistical analysis

Previous studies have reported a success rate of 95% at 6 months for Edwards MC3 [21,22]. This study was designed to assess the non-inferiority of PM-AR-T, with a predefined non-inferiority margin of −10% regarding 6-month postoperative success rate of tricuspid valve repair [23]. Considering a one-sided α of 0.025, a power of 0.8 and a dropout rate of 10%, we aimed to enroll 82 patients per group, resulting in a total of 164 TR patients.

The primary efficacy endpoint was assessed in both the full analysis set (FAS) and the per-protocol set (PPS). Secondary efficacy endpoints were assessed in the FAS. The FAS included all randomized patients who underwent surgical

implantation of a device. The PPS comprised patients who adhered rigorously to the trial protocol, including treatment compliance, the availability of requisite measurements for primary endpoint assessment, and the absence of substantial protocol violations. Safety evaluations were performed in the safety set (SS), which encompassed all randomized patients receiving the device implant, with a detailed recording of safety-related data post-implantation. Quantitative data were assessed for normality using the Kolmogorov-Smirnov test.

To address missing data for the primary endpoint, in cases without primary endpoint data and 12-month follow-up, the absent primary endpoint data were inferred as "unsuccessful tricuspid valve annuloplasty ring repair at six months postoperatively." Conversely, in cases missing primary endpoint data but possessing 12-month follow-up data, echocardiographic outcomes at 12 months substituted the missing primary endpoint data in the analysis.

The Newcombe method was used to compute the two-sided 95% confidence interval (CI) for the difference in success rate for tricuspid valve annuloplasty ring repair at six months postoperatively between the PM-AR-T and Edwards MC3 groups. With a predefined non-inferiority margin of −10%, this approach facilitated the assessment of whether the PM-AR-T was non-inferior to Edwards MC3 in terms of performance. Recognizing the potential effects of both the participating center and regurgitation severity on repair success rate, the corrected Mantel-Haenszel (CMH) method was applied to determine the two-sided 95% CI for the adjusted rate difference. Logistic regression models further elucidated odds ratios (ORs) and 95% CIs for repair success rate among groups, incorporating center and regurgitation severity as covariates.

## Results

### Baseline characteristics

A total of 166 patients were randomized, and one patient from each group was excluded for non-participation in the surgical procedure following randomization. Therefore, 164 patients were successfully undergoing valve annuloplasty ring implantation, including 83 and 81 in the PM-AR-T and Edwards MC3 groups, respectively. Subsequent follow-up at six months post-surgery revealed 3 versus 1 patients were lost to follow-up in the PM-AR-T and the Edwards MC3 group. Consequently, the FAS, PPS and SS included 83, 80 and 83 patients in the PM-AR-T group, while 81, 80 and 81 patients in the Edwards MC3 group (Fig 1).

Table 1 presents a comparative overview of baseline characteristics. The average age was 55.70±10.63 and 55.40±11.33 years, with 38.6% and 37.0% males in the PM-AR-T and Edwards MC3 groups. The prevalence rates of coronary artery disease, hypertension, and hyperlipidemia were comparable between the two groups (all P>0.05). All patients in both groups had tricuspid valve insufficiency, with similar proportions of other valve diseases. The NYHA functional classification was closely matched between the two groups, with most cases classified as class III. The degrees of regurgitation were evenly distributed among mild, moderate, and severe cases in both groups.

### Surgery-related data

The mean operative durations were 5.33±1.62 and 4.90±1.20 hours in the PM-AR-T and Edwards MC3 groups, respectively. Concomitant procedures were performed in 98.8% and 100% of all patients in the PM-AR-T and Edwards MC3 groups, respectively. The proportion of each type of annuloplasty ring usage was comparable between the two groups (P=0.260) (Table 2). The primary surgical access was through the right atrium (100% in the two groups).

### Primary endpoint

In the FAS (Table 3), the PM-AR-T and Edwards MC3 groups had success rates of 92.8% (77/83 patients) and 93.8% (76/81 patients), respectively. The intergroup difference, determined by the Newcombe method, was −1.1% (95% CI, −9.5 to 7.4), confirming non-inferiority between the two groups. Subsequent adjustments for center and severity of

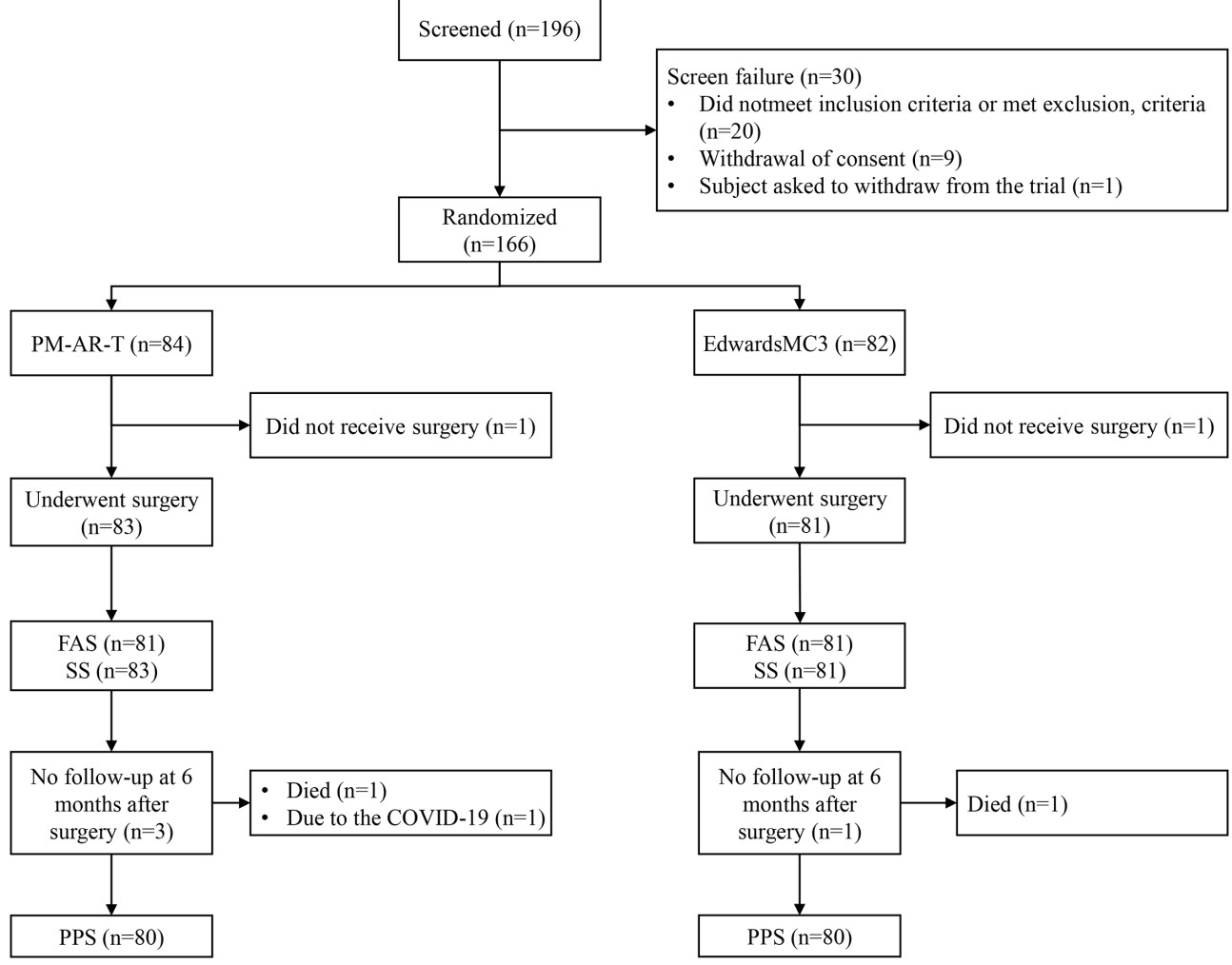

**Fig 1. Study flowchart.** FAS, full analysis set; SS, safety analysis set; PPS, per-protocol analysis set.

regurgitation, employing the CMH method, adjusted this difference to −1.2% (95% CI, −9.3 to 6.8). Subgroup analysis further reinforced these findings, revealing no significant differences in success rate across different severities of regurgitation in both groups.

Parallel results were observed in the PPS analysis, where the PM-AR-T group achieved a success rate of 96.3% (77/80 patients) versus 95.0% (76/80 patients) in the Edwards MC3 group. The Newcombe method suggested an intergroup difference of 1.3% (95% CI, −6.1 to 8.8), again supporting the non-inferiority finding. Adjustments for center and regurgitation severity mirrored the initial analysis, yielding a slight difference of 1.5% in success rate (95% CI, −5.2 to 8.3). Subgroup analysis also underscored a consistent level of repair efficacy between the two groups, irrespective of regurgitation severity.

### Secondary endpoints

Secondary endpoints are presented in Table 4. There were no statistically significant differences in valve regurgitation severity and echocardiographic parameters between the PM-AR-T and Edwards MC3 groups at 30 days, 3 months,

**Table 1. Baseline characteristics of patients.**

| Variable | PM-AR-T Group (n = 83) | Edwards MC3 Group (n = 81) | P |
|---|---|---|---|
| Age (years) | 55.70 ± 10.63 | 55.40 ± 11.33 | 0.865 |
| Gender, n (%) | | | 0.841 |
| Male | 32 (38.6) | 30 (37.0) | |
| Female | 51 (61.4) | 51 (63.0) | |
| Coronary artery disease, n (%) | 18 (21.7) | 13 (16.0) | 0.357 |
| Hypertension, n (%) | 16 (19.3) | 19 (23.5) | 0.514 |
| Hyperlipidemia, n (%) | 6 (7.2) | 5 (6.2) | 0.787 |
| Atrial fibrillation, n (%) | 60 (72.3) | 47 (58.0) | 0.055 |
| Angina pectoris, n (%) | 4 (4.8) | 0 | 0.121 |
| Valve disease, n (%) | | | |
| Aortic valve insufficiency or stenosis | 44 (53.0) | 40 (49.4) | 0.327 |
| Mitral valve insufficiency or stenosis | 78 (94.0) | 73 (90.1) | 0.480 |
| Tricuspid valve insufficiency | 83 (100.0) | 81 (100.0) | NA |
| Pulmonary valve insufficiency or stenosis | 6 (7.2) | 6 (7.4) | >0.999 |
| History of PCI, CABG, n (%) | 1 (1.2) | 1 (1.2) | >0.999 |
| NYHA functional classification, n (%) | | | 0.537 |
| I | 1 (1.2) | 2 (2.5) | |
| II | 26 (31.3) | 20 (24.7) | |
| III | 56 (67.5) | 59 (72.8) | |
| IV | 0 | 0 | |
| Degree of regurgitation, n (%) | | | 0.857 |
| Mild | 7 (8.4) | 5 (6.2) | |
| Moderate | 33 (39.8) | 33 (40.7) | |
| Severe | 43 (51.8) | 43 (53.1) | |
| Echocardiographic parameters | | | |
| LVESD (mm) | | | 0.477 |
| Missing, n | 1 | 1 | |
| Mean±SD | 35.14 ± 7.51 | 34.25 ± 8.25 | |
| LVEDD (mm), Mean±SD | 49.64 ± 8.74 | 49.33 ± 9.92 | 0.835 |
| Right atrial diameter at end-systole (mm), Mean ± SD | 42.67 ± 7.95 | 42.41 ± 10.14 | 0.858 |
| RV (mm) | | | 0.503 |
| Missing, n | 0 | 1 | |
| Mean±SD | 35.15 ± 8.07 | 34.27 ± 8.70 | |
| EF (%), Mean±SD | 59.13 ± 8.86 | 59.13 ± 9.87 | >0.999 |
| Absolute value of regurgitant flow area (cm$^2$) | | | 0.743 |
| Missing, n | 1 | 2 | |
| Mean±SD | 7.70 ± 4.11 | 7.93 ± 4.84 | |
| Annulus diameter (mm) | | | 0.373 |
| Missing, n | 30 | 29 | 0.774 |
| Regurgitant jet extends ≤1 cm from annulus, n (%) | 24 (45.3) | 25 (48.1) | |
| Regurgitant jet extends >1 cm from annulus, n (%) | 29 (54.7) | 27 (51.9) | |

LVEDD, left ventricular end-diastolic diameter; LVESD, left ventricular end-systolic diameter; EF, ejection fraction; NYHA, New York Heart Association; RV, right ventricular end-diastolic diameter; PCI, percutaneous coronary intervention; CABG, coronary artery bypass grafting; SD, standard deviation.

**Table 2. Surgical data of the patients.**

| Variable | PM-AR-T Group (n=83) | Edwards MC3 Group (n=81) | P |
|---|---|---|---|
| Surgical time (h) | 5.33±1.62 | 4.90±1.20 | 0.058 |
| Patients with additional surgeries, n (%) | 82 (98.8) | 81 (100.0) | >0.999 |
| Mitral valve repair | 21 (25.3) | 26 (32.1) | |
| Mitral valve replacement | 55 (66.3) | 40 (49.4) | |
| Cardiac radiofrequency ablation | 45 (54.2) | 23 (28.4) | |
| Aortic valve replacement | 14 (16.9) | 11 (13.6) | |
| Atrial septal defect repair | 8 (9.6) | 6 (7.4) | |
| Patent foramen ovale closure | 1 (1.2) | 2 (2.5) | |
| Left atrial surgery | 23 (27.7) | 12 (14.8) | |
| Coronary artery bypass grafting | 3 (3.6) | 4 (4.9) | |
| Left atrial appendage closure | 5 (6.0) | 9 (11.1) | |
| Other | 28 (33.7) | 26 (32.1) | |
| Suturing technique | | | >0.999 |
| Continuous suturing method, n (%) | 3 (3.6) | 2 (2.5) | |
| Interrupted suturing method, n (%) | 80 (96.4) | 79 (97.5) | |
| Valve ring measurement (mm), Mean±SD | 29.0±1.44 | 28.8±1.22 | 0.240 |
| Annuloplasty ring size (mm), Mean±SD | 29.0±1.44 | 28.8±1.21 | 0.289 |
| Annuloplasty ring size | | | 0.260 |
| M26 | 2 (2.4) | 0 | |
| M28 | 46 (55.4) | 54 (66.7) | |
| M30 | 26 (31.3) | 22 (27.2) | |
| M32 | 9 (10.8) | 5 (6.2) | |

6 months, and 12 months postoperatively (all P>0.05). Specifically, the proportions of patients without regurgitation were 30.4% and 27.8% in the PM-AR-T and Edwards MC3 groups at 12 months, respectively (P=0.705). The mean right atrial dimensions remained consistent between the two groups across all time points, measuring 36.92±5.81 mm and 37.29±5.89 mm, respectively, in the PM-AR-T and Edwards MC3 groups at 30 days (P=0.687) and narrowing to 35.88±6.02 mm and 36.14±5.73 mm, respectively, at 12 months (P=0.781). Right ventricular dimension and ejection fraction also showed no significant differences between the two groups during the follow-up period. At 30 days, right ventricular end-diastolic diameter (RV) values were 31.74±6.37 mm and 31.02±7.19 mm in the PM-AR-T and Edwards MC3 groups, respectively (P=0.502), with minimal changes at 12 months (30.43±6.18 mm versus 29.46±5.90 mm, P=0.314). Ejection fraction (EF) percentages paralleled this trend, with no statistical differences detected. The improvement to NYHA functional class I status by 12 months was detected in 46.2% and 45.6% of the PM-AR-T and Edwards MC3 groups (P=0.893). No patients were classified as NYHA functional class IV postoperatively.

Additionally, both groups exhibited a 100% immediate postoperative technical success rate, with no patients requiring repeat surgery for TR.

## Safety profiles

Throughout the 12-month period, no device-related SAEs, cardiovascular deaths, major bleeding events, severe structural damage, infective endocarditis, or thromboembolic events were detected (Table 5).

TEAEs occurred in 76 (91.6%) and 76 (93.8%) patients of the PM-AR-T and Edwards MC3 groups, respectively. In the PM-AR-T group, two patients (2.4%) experienced device-related TEAEs (one case each of pericardial effusion and organ failure), versus two patients (2.5%) in the Edwards MC3 group (one case each of second-degree atrioventricular block

**Table 3. Repair success rates of the valve annuloplasty rings at 6 months post-surgery.**

| | PM-AR-T Group | Edwards MC3 Group |
|---|---|---|
| FAS | | |
| Success rate, % | 92.8% (77/83) | 93.8% (76/81) |
| Rate difference, 95% CI | −1.1% (−9.5%, 7.4%) | |
| Rate difference, 95% CI[a] | −1.2% (−9.3%, 6.8%) | |
| Subgroup analysis | | |
| Mild regurgitation, n | 7 | 5 |
| Success rate, n (%) | 7 (100.0) | 5 (100.0) |
| Moderate regurgitation, n | 33 | 33 |
| Success rate, n (%) | 32 (97.0) | 31 (93.9) |
| Severe regurgitation, n | 43 | 43 |
| Success rate, n (%) | 38 (88.4) | 40 (93.0) |
| PPS | | |
| Success rate, % | 96.3% (77/80) | 95.0% (76/80) |
| Rate difference, 95% CI | 1.3% (−6.1%, 8.8%) | |
| Rate difference, 95% CI[a] | 1.5% (−5.2%, 8.3%) | |
| Subgroup analysis | | |
| Mild regurgitation, n | 7 | 5 |
| Success rate, n (%) | 7 (100.0) | 5 (100.0) |
| Moderate regurgitation, n | 32 | 33 |
| Success rate, n (%) | 32 (100.0) | 31 (93.9) |
| Severe regurgitation, n | 41 | 42 |
| Success rate, n (%) | 38 (92.7) | 40 (95.2) |

FAS, full analysis set; PPS, per-protocol set; a, adjusted for center and degree of regurgitation.

and atrial fibrillation with complete atrioventricular block). A total of 70 patients (84.3%) in the PM-AR-T group reported surgery-related TEAEs, with the most common being inflammation (47.0%), anemia (36.1%), and decreased platelet count (26.5%), while 67 patients (82.7%) in the Edwards MC3 group, including inflammation (45.7%), anemia (37.0%), and decreased platelet count (30.9%) (Table 6).

SAEs occurred in 28 (33.7%) and 18 (22.2%) patients in the PM-AR-T and Edwards MC3 groups, respectively, with one device-related SAE observed in the PM-AR-T group and none in the Edwards MC3 group. Overall, 11 (13.3%) and 7 (8.6%) patients had surgery-related SAEs in the PM-AR-T and Edwards MC3 groups, respectively. No TEAEs led to treatment discontinuation in either group. Notably, there were two deaths (2.4%) in the PM-AR-T group, with one each due to serious co-morbidities and infectious shock causing multiorgan failure. Similarly, the Edwards MC3 group had two deaths (2.5%), with one each due to acute kidney injury/subdural hemorrhage and large cerebral infarction in the left cerebral hemisphere (Table 5).

## Discussion

In the quest to enhance the management of TR, this study performed a comparative evaluation of two distinct annuloplasty rings: PM-AR-T and Edwards MC3. The findings established the non-inferiority of the PM-AR-T versus Edwards MC3 in terms of success rate (92.8% and 93.8%, respectively) within the FAS, revealing an intergroup variance of −1.1% (95% CI, −9.5% to 7.4%). This parity in performance was further echoed in PPS analysis, underscoring the consistency of the PM-AR-T ring's efficacy. Moreover, the safety assessment of both annuloplasty devices revealed comparable profiles,

**Table 4. Analysis of valve regurgitation, echocardiographic parameters, and NYHA functional classification postoperatively.**

| | 30 days postoperatively | | | 3 months postoperatively | | | 6 months postoperatively | | | 12 months postoperatively | | |
|---|---|---|---|---|---|---|---|---|---|---|---|---|
| | PM-AR-T Group | Edwards MC3 Group | P | PM-AR-T Group | Edwards MC3 Group | P | PM-AR-T Group | Edwards MC3 Group | P | PM-AR-T Group | Edwards MC3 Group | P |
| Valve regurgitation severity | | | 0.432 | | | 0.067 | | | 0.243 | | | 0.705 |
| N (Miss) | 81 (2) | 80 (1) | | 75 (8) | 75 (6) | | 79 (4) | 79 (2) | | 79 (4) | 79 (2) | |
| None, n (%) | 16 (19.8) | 26 (32.5) | | 18 (24.0) | 23 (30.7) | | 18 (22.8) | 21 (26.6) | | 24 (30.4) | 22 (27.8) | |
| Trace, n (%) | 16 (19.8) | 11 (13.8) | | 10 (13.3) | 19 (25.3) | | 13 (16.5) | 22 (27.8) | | 17 (21.5) | 13 (16.5) | |
| Mild, n (%) | 38 (46.9) | 32 (40.0) | | 38 (50.7) | 21 (28.0) | | 36 (45.6) | 23 (29.1) | | 32 (40.5) | 33 (41.8) | |
| Moderate, n (%) | 7 (8.6) | 7 (8.8) | | 4 (5.3) | 6 (8.0) | | 9 (11.4) | 10 (12.7) | | 4 (5.1) | 8 (10.1) | |
| Severe, n (%) | 4 (4.9) | 4 (5.0) | | 5 (6.7) | 6 (8.0) | | 3 (3.8) | 3 (3.8) | | 2 (2.5) | 3 (3.8) | |
| **Echocardiographic parameters** | | | | | | | | | | | | |
| Right atrial dimension (mm) | | | | | | | | | | | | |
| N (Miss) | 80 (3) | 80 (1) | 0.687 | 74 (9) | 74 (7) | 0.595 | 79 (4) | 78 (3) | 0.838 | 79 (4) | 79 (2) | 0.781 |
| Mean±SD | 36.92±5.81 | 37.29±5.89 | | 35.92±5.35 | 35.47±4.92 | | 36.40±5.63 | 36.59±5.95 | | 35.88±6.02 | 36.14±5.73 | |
| RV (mm) | | | | | | | | | | | | |
| N (Miss) | 80 (3) | 80 (1) | 0.502 | 74 (9) | 74 (7) | 0.579 | 78 (5) | 78 (3) | 0.590 | 79 (4) | 79 (2) | 0.314 |
| Mean±SD | 31.74±6.37 | 31.02±7.19 | | 29.77±7.39 | 29.10±7.22 | | 29.88±6.20 | 29.27±7.84 | | 30.43±6.18 | 29.46±5.90 | |
| EF (%) | | | | | | | | | | | | |
| N (Miss) | 81 (2) | 80 (1) | 0.901 | 75 (8) | 75 (6) | 0.344 | 78 (5) | 79 (2) | 0.515 | 78 (5) | 79 (2) | 0.580 |
| Mean±SD | 58.91±7.99 | 59.08±8.73 | | 59.60±9.34 | 60.94±7.87 | | 59.88±8.37 | 60.75±8.39 | | 61.07±7.10 | 61.75±8.31 | |
| Annular long diameter (mm) | | | | | | | | | | | | |
| N (Miss) | 80 (3) | 79 (2) | 0.921 | 74 (9) | 73 (8) | 0.404 | 79 (4) | 79 (2) | 0.574 | 79 (4) | 80 (1) | 0.734 |
| Mean±SD | 25.960±3.34 | 26.017±3.80 | | 25.631±3.21 | 26.097±3.55 | | 25.945±2.88 | 25.644±3.79 | | 25.895±3.14 | 26.094±4.16 | |
| Annular short diameter | | | | | | | | | | | | |
| N (Miss) | 80 (3) | 79 (2) | 0.659 | 74 (9) | 73 (8) | 0.822 | 79 (4) | 79 (2) | 0.396 | 79 (4) | 80 (1) | 0.678 |
| Mean±SD | 18.655±3.62 | 18.381±4.18 | | 18.184±3.18 | 18.315±3.82 | | 18.669±3.42 | 18.214±3.30 | | 18.571±3.03 | 18.355±3.52 | |
| Narrowest width of regurgitant jet (Jet Width) (mm) | | | | | | | | | | | | |
| N (Miss) | 62 (21) | 58 (23) | 0.165 | 57 (26) | 54 (27) | 0.663 | 52 (31) | 57 (24) | 0.941 | 63 (20) | 66 (15) | 0.608 |
| Regurgitant jet extension to right Atrium ≤1 cm, n (%) | 11 (17.7) | 19 (32.8) | | 14 (24.6) | 17 (31.5) | | 10 (19.2) | 12 (21.1) | | 22 (34.9) | 20 (30.3) | |
| Regurgitant jet extension to right Atrium >1 cm, n (%) | 30 (48.4) | 23 (39.7) | | 26 (45.6) | 24 (44.4) | | 25 (48.1) | 28 (49.1) | | 25 (39.7) | 24 (36.4) | |
| No regurgitant jet in right atrium, n (%) | 21 (33.9) | 16 (27.6) | | 17 (29.8) | 13 (24.1) | | 17 (32.7) | 17 (29.8) | | 16 (25.4) | 22 (33.3) | |

(Continued)

**Table 4.** (Continued)

| | 30 days postoperatively | | | 3 months postoperatively | | | 6 months postoperatively | | | 12 months postoperatively | |
|---|---|---|---|---|---|---|---|---|---|---|---|
| Absolute value of regurgitant flow area (cm²) | | | | | | | | | | | |
| N (Miss) | 74 (9) | 74 (7) | 0.854 | 72 (11) | 68 (13) | 0.995 | 74 (9) | 71 (10) | 0.878 | 73 (10) | 72 (9) | 0.896 |
| Mean±SD | 1.648±2.20 | 1.713±2.11 | | 1.571±1.92 | 1.569±2.29 | | 1.691±2.08 | 1.637±2.11 | | 1.461±2.31 | 1.505±1.77 | |
| NYHA functional classification | | | | | | | | | | | |
| N (Miss) | 82 (1) | 81 (0) | 0.584 | 76 (7) | 75 (6) | 0.899 | 78 (5) | 78 (3) | 0.281 | 78 (5) | 79 (2) | 0.893 |
| I, n (%) | 2 (2.4) | 3 (3.7) | | 46 (60.5) | 41 (54.7) | | 50 (64.1) | 38 (48.7) | | 36 (46.2) | 36 (45.6) | |
| II, n (%) | 42 (51.2) | 35 (43.2) | | 24 (31.6) | 28 (37.3) | | 22 (28.2) | 30 (38.5) | | 24 (30.8) | 28 (35.4) | |
| III, n (%) | 38 (46.3) | 43 (53.1) | | 4 (5.3) | 4 (5.3) | | 2 (2.6) | 4 (5.1) | | 2 (2.6) | 2 (2.5) | |
| IV, n (%) | 0 | 0 | | 0 (0) | 0 (0) | | 0 (0) | 0 (0) | | 0 (0) | 0 (0) | |
| Not applicable, n (%) | 0 | 0 | | 2 (2.6) | 2 (2.7) | | 4 (5.1) | 6 (7.7) | | 16 (20.5) | 13 (16.5) | |

EF, ejection fraction; RV, right ventricular end-diastolic diameter; NYHA, New York Heart Association.

**Table 5. Postoperative adverse events.**

| | PM-AR-T Group (n = 83) | Edwards MC3 Group (n = 81) |
|---|---|---|
| **Secondary safety endpoints** | | |
| Device-related SAE | 0 | 0 |
| Device-related cardiovascular mortality, n (%) | 0 | 0 |
| Device-related major bleeding, n (%) | 0 | 0 |
| Device-related severe structural or other body structure damage, n (%) | 0 | 0 |
| Device-related infective endocarditis, n (%) | 0 | 0 |
| Device-related thromboembolic events, n (%) | 0 | 0 |
| All-cause mortality | 2 (2.4) | 2 (2.5) |
| **Other Adverse Events** | | |
| TEAE | 76 (91.6) | 76 (93.8) |
| Device-related TEAE, n (%) | 2 (2.4) | 2 (2.5) |
| Surgery-related TEAE, n (%) | 70 (84.3) | 67 (82.7) |
| TEAE leading to trial discontinuation, n (%) | 0 | 0 |
| Device-related TEAE leading to trial discontinuation, n (%) | 0 | 0 |
| Surgery-related TEAE leading to trial discontinuation, n (%) | 0 | 0 |
| TEAE leading to death, n (%) | 2 (2.4) | 2 (2.5) |
| Device-related TEAE leading to death, n (%) | 1 (1.2) | 0 |
| Surgery-related TEAE leading to death, n (%) | 1 (1.2) | 1 (1.2) |
| SAE | 28 (33.7) | 18 (22.2) |
| Device-related SAE, n (%) | 1 (1.2) | 0 |
| Surgery-related SAE, n (%) | 11 (13.3) | 7 (8.6) |

TEAE, treatment-emergent adverse event; SAE, serious adverse events.

with the PM-AR-T ring demonstrating a controllable and satisfactory safety profile. These findings confirmed the PM-AR-T ring as a viable, safe option in the pantheon of tricuspid valve repair tools.

The Edwards MC3 annuloplasty ring is considered a cornerstone of TR management. Some reports highlighted the ring's capacity to significantly reduce TR severity post-implantation [11], with these improvements being maintained over time [12,24]. Such outcomes underscore the role of the MC3 ring in remodeling the tricuspid valve annulus while preserving physiological function, a critical aspect in mitigating the recurrence of TR. Some studies revealed that patients treated with the MC3 ring exhibit lower rates of recurrent TR over time, emphasizing the importance of ring rigidity in achieving lasting valve competence [13,25]. Longitudinal studies focusing on long-term outcomes of tricuspid annuloplasty with 3D-shaped rings, including MC3, have further validated their effectiveness and durability [26,27]. The recurrence of TR was relatively low, confirming the sustained performance of the MC3 ring in tricuspid valve repair.

This study expands upon this body of evidence by integrating the PM-AR-T semi-rigid annuloplasty ring into the comparative framework. The success rates of valve annuloplasty at 6 months postoperatively between the PM-AR-T and Edwards MC3 groups were remarkably similar (92.8% vs. 93.8%), indicating no significant difference in efficacy. This parity persisted across various levels of regurgitation severity and was consistent in both FAS and PPS. Moreover, this study examining secondary efficacy endpoints, including improvements in echocardiographic parameters and NYHA functional class, revealed no discernible differences between the two groups, further confirming the comparable performance of the PM-AR-T ring. Notably, the absence of reoperation for TR in both groups highlights the

**Table 6. Most common TEAEs postoperatively related to device and surgery.**

| | PM-AR-T Group (n = 83) | Edwards MC3 Group (n = 81) |
|---|---|---|
| Device-related TEAE, n (%) | 2 (2.4) | 2 (2.5) |
| Pericardial effusion | 1 (1.2) | 0 |
| Organ failure | 1 (1.2) | 0 |
| Second-degree atrioventricular block | 0 | 1 (1.2) |
| Atrial fibrillation | 0 | 1 (1.2) |
| Complete atrioventricular block | 0 | 1 (1.2) |
| Surgery-related TEAE (≥10% in either group), n (%) | 70 (84.3) | 67 (82.7) |
| Inflammation | 39 (47.0) | 37 (45.7) |
| Anemia | 30 (36.1) | 30 (37.0) |
| Decreased platelet count | 22 (26.5) | 25 (30.9) |
| Pleural effusion | 22 (26.5) | 20 (24.7) |
| Pain | 16 (19.3) | 17 (21.0) |
| Fever | 14 (16.9) | 15 (18.5) |
| Hypoproteinemia | 11 (13.3) | 8 (9.9) |
| Pulmonary inflammation | 10 (12.0) | 16 (19.8) |
| Myocardial injury | 10 (12.0) | 15 (18.5) |
| Myocardial reperfusion injury | 10 (12.0) | 8 (9.9) |
| Hypovolemia | 9 (10.8) | 7 (8.6) |
| Abnormal liver function | 9 (10.8) | 4 (4.9) |
| Infectious pneumonia | 9 (10.8) | 6 (7.4) |
| Vomiting | 8 (9.6) | 10 (12.3) |

TEAE, treatment-emergent adverse event.

long-term efficacy and reliability of these interventions. These findings, therefore, not only confirm the established efficacy of the Edwards MC3 ring but also present the PM-AR-T semi-rigid ring as a viable, non-inferior alternative for tricuspid valve repair.

Previous studies of the Edwards MC3 ring have consistently demonstrated a reassuring safety profile across diverse cohorts of patients with TR [11]. Furthermore, comparative analyses have revealed the Edwards MC3 ring to exhibit comparable early mortality and complication rates to other annuloplasty devices in TR [13,25]. Building on this substantial body of evidence, the current study performed a comparative analysis of the PM-AR-T semi-rigid annuloplasty ring, confirming a favorable safety profile similar to the Edwards MC3 ring. Throughout the 12-month observation period, no device-related SAEs, cardiovascular deaths, major bleeding events, severe structural damage, infective endocarditis, or thromboembolic events attributable to either device were recorded. The incidence of device-related AEs resulting in death was minimal and comparable between the two groups, further supporting the safety and reliability of the PM-AR-T ring in the clinic. Moreover, the current findings on the incidence of TEAEs and surgery-related SAEs are consistent with previous clinical experiences with the Edwards MC3 ring. The absence of significant differences in the rates of device-related AEs between the PM-AR-T and Edwards MC3 groups and the lack of TEAEs resulting in trial discontinuation consolidate the notion that PM-AR-T is a viable and safe option for tricuspid valve repair.

One of the primary limitations of this study was the duration of follow-up, which was confined to a 12-month period. This temporal limitation precludes a comprehensive assessment of long-term survival outcomes as well as the sustained benefits of improved cardiac function postoperatively. Given the chronic nature of TR and the potential for late-onset

complications or recurrence, it is imperative that future studies extend beyond the initial postoperative year. Therefore, larger prospective studies not only applying a more extended follow-up period but also including real-world settings are required. Furthermore, post hoc analyses of clinical trial populations can be valuable in identifying predictors of long-term outcomes. A recent study by Wedin et al. used machine learning techniques to stratify risk in patients with functional mitral regurgitation undergoing TEER, with primary outcomes of cardiovascular death or hospitalization for heart failure (HF) at one year [28]. The retrospective analysis of data from this study population is expected, aiming to explore factors influencing other clinically meaningful outcomes after surgery, and similarly, retrospective analysis of data from this study may help identify risk factors associated with TR recurrence, functional deterioration, or the need for reintervention, thereby informing patient selection and long-term management strategies in future research.

## Conclusion

This study provides evidence that in patients with TR, the nickel-titanium-based PM-AR-T tricuspid semi-rigid annuloplasty ring provides an effective solution for improving regurgitation outcomes. Demonstrating non-inferiority to the well-established Edwards MC3 tricuspid annuloplasty ring, the PM-AR-T ring not only achieves comparable efficacy in the management of TR but also maintains a commendable safety profile.

## Supporting information

**S1 Checklist. CONSORT 2010 checklist of information to include when reporting a randomised trial.**
(DOC)

**S2 Checklist. PLOSOne Human Subjects Research Checklist.**
(DOCX)

**S1 Section. PM-AR-T introduction.**
(DOCX)

**S1 Table. Comparative specifications of multi-model devices.**
(DOCX)

**S1 Fig. Structural components of Permedos PM-AR-T annuloplasty ring device.** A. Device assembly schematic. B. Composite schematic: annuloplasty ring with holder. C. Holder handle assembly schematic. D. Annuloplasty ring.
(TIF)

**S2 Fig. Comparative illustration of the Edwards MC3 versus PM-AR-T tricuspid annuloplasty rings.** A. Overview of Edwards MC3 and PM-AR-T tricuspid annuloplasty rings. B. Cross-section of the Edwards MC3 tricuspid annuloplasty ring. C. Cross-section of the PM-AR-T tricuspid annuloplasty ring. D. Stiffness distribution map of Edwards MC3. E. Stiffness distribution map of the PM-AR-T and an enlarged view of its hollow-patterned tube.
(TIF)

**S1 File. Data presented in this paper.**
(XLSX)

**S2 File. Study Protocol.**
(PDF)

## Acknowledgments

None

## Author contributions

**Conceptualization:** Zhenjun Xu, Jie Li, Kunsheng Li, Dongjing Wang, Jun Pan.

**Data curation:** Zhenjun Xu, Jie Li, Wanzi Xu, Qiuyan Zong, Wenjie Ji, Yin Xu, Yunyan Su, Dongjing Wang, Jun Pan.

**Formal analysis:** Zhenjun Xu, Jie Li, Dongjing Wang, Jun Pan.

**Methodology:** Zhenjun Xu, Jie Li, Dongjing Wang, Jun Pan.

**Resources:** Wanzi Xu, Qiuyan Zong, Yin Xu, Yunyan Su.

**Writing – original draft:** Zhenjun Xu, Jie Li, Jun Pan.

**Writing – review & editing:** Zhenjun Xu, Jie Li, Wenjie Ji, Yin Xu, Yunyan Su, Dongjing Wang, Jun Pan.

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
