## [Decision Letter · Decision Letter 0]

13 Jun 2025

Dear Dr. Wang,

Thank you for submitting your manuscript to PLOS ONE. After careful consideration, we feel that it has merit but does not fully meet PLOS ONE’s publication criteria as it currently stands. Therefore, we invite you to submit a revised version of the manuscript that addresses the points raised during the review process.

We look forward to receiving your revised manuscript.

Kind regards,

Kamal Sharma

Academic Editor

PLOS ONE

Journal Requirements:

“Beijing Permed Biomedical Engineering Co., Ltd.”

3. We note that your Data Availability Statement is currently as follows: All relevant data are within the manuscript and its Supporting Information files

5. We note you have included a table to which you do not refer in the text of your manuscript. Please ensure that you refer to Table 4 in your text; if accepted, production will need this reference to link the reader to the Table.

Reviewers' comments:

Reviewer's Responses to Questions

**Comments to the Author**

1. Is the manuscript technically sound, and do the data support the conclusions?

Reviewer #1: Yes

Reviewer #2: Yes

2. Has the statistical analysis been performed appropriately and rigorously?

Reviewer #1: Yes

Reviewer #2: Yes

3. Have the authors made all data underlying the findings in their manuscript fully available?

Reviewer #1: Yes

Reviewer #2: No

4. Is the manuscript presented in an intelligible fashion and written in standard English?

Reviewer #1: Yes

Reviewer #2: Yes

Reviewer #1: Interesting paper.

Some issues should be addressed

164 patients in 20 centers means a small number of patients for center. How were centers selected?

definition of success rate should be added

due to small number of patients, normal distribution shoulb be checked for

clustering with ML has recently been proposes for a risk stratification for MR. Please quote and comment on PMID 40395430

Reviewer #2: The RCT is designed to evaluate the safety and efficacy of PM-AR-T.

Following are comments to be addressed.

Comment 1: The manuscript designed with 1:1 randomization how ever there is no information How patient were randomized between investigational device and control.

Comment 2: Introduction section needs more strengthening e.g. side-by-side technical specification and visual comparison of design and pointing features of investigational device and control. Summary of prior results of investigational device.

Comment 3: Reference 15 coated animal study of test device, however detailed results in manuscript body would be helpful for readers. Moreover, providing DOI in reference section would help to read detailed results about test device.

Comment 4: Any COVID related assessment on outcome would be helpful.

Comment 5: A brief information of data evaluation, data safety monitoring committee and steering committee would be useful for readers.

Comment5: Raw data sheet for statistical calculation should be available as supplementary data sheet.

**Do you want your identity to be public for this peer review?** For information about this choice, including consent withdrawal, please see our Privacy Policy

Reviewer #1: **Yes: ** Fabrizio D'Ascenzo

Reviewer #2: No

---

## [Author Response · Author response to Decision Letter 1]

29 Jul 2025

Due to Reviewer 2's inquiry, we have added important results from unpublished preclinical (animal) studies in our rebuttal letter. As the tables could not be directly copied into the manuscript revision system, we have uploaded the entire rebuttal document named “Rebutal letter-07.24 revised” for the editor and reviewers to review.

---

## [Decision Letter · Decision Letter 1]

22 Sep 2025

Efficacy and Safety of PM-AR-T versus Edwards MC3 Rings in Tricuspid Regurgitation: A Non-inferiority, Randomized Controlled Trial

PONE-D-25-24627R1

Dear Dr. Wang,

We’re pleased to inform you that your manuscript has been judged scientifically suitable for publication and will be formally accepted for publication once it meets all outstanding technical requirements.

Kind regards,

Kamal Sharma

Academic Editor

PLOS ONE

Additional Editor Comments (optional):

Reviewer #1:

Reviewer #2:

Reviewers' comments:

Reviewer's Responses to Questions

**Comments to the Author**

Reviewer #1: (No Response)

Reviewer #2: All comments have been addressed

2. Is the manuscript technically sound, and do the data support the conclusions?

Reviewer #1: Yes

Reviewer #2: Yes

3. Has the statistical analysis been performed appropriately and rigorously?

Reviewer #1: Yes

Reviewer #2: Yes

4. Have the authors made all data underlying the findings in their manuscript fully available?

Reviewer #1: Yes

Reviewer #2: Yes

5. Is the manuscript presented in an intelligible fashion and written in standard English?

Reviewer #1: Yes

Reviewer #2: Yes

Reviewer #1: Interesing and well condeucted study. Ony Minor issues

1) in statistical anaysis authors should check for normality

2) In the multivariate analysis I would ass also gender, degree of RV dynsfunction and anulis dilatation

3) results: p for difference should be added

4)recently clustering with ML has provided interesting results. do authors think it may help to stratify risk also in this setting? quote on PMID: 40395430

Reviewer #2: All comments have been address to the article titled "Efficacy and Safety of PM-AR-T versus Edwards MC3 Rings in Tricuspid Regurgitation: A Non-inferiority, Randomized Controlled Trial". It can now be accepted.

**Do you want your identity to be public for this peer review?** For information about this choice, including consent withdrawal, please see our Privacy Policy

Reviewer #1: **Yes: ** Fabrizio D'Ascenzo

Reviewer #2: No

---

## [Editor Report · Acceptance letter]

PONE-D-25-24627R1

PLOS ONE

Dear Dr. Wang,

I'm pleased to inform you that your manuscript has been deemed suitable for publication in PLOS ONE. Congratulations! Your manuscript is now being handed over to our production team.

Kind regards,

on behalf of

Dr. Kamal Sharma

Academic Editor

PLOS ONE